# Bayesian Weighted Sums: A Flexible Approach to Estimate Summed Mixture Effects

**DOI:** 10.3390/ijerph18041373

**Published:** 2021-02-03

**Authors:** Ghassan B. Hamra, Richard F. Maclehose, Lisa Croen, Elizabeth M. Kauffman, Craig Newschaffer

**Affiliations:** 1Department of Epidemiology, Johns Hopkins Bloomberg School of Public Health, Baltimore, MD 21205, USA; newschaffer@psu.edu; 2Division of Epidemiology and Community Health, University of Minnesota, Minneapolis, MN 55455, USA; macl0029@umn.edu; 3Division of Research, Kaiser Permanente, Oakland, CA 94612, USA; Lisa.A.Croen@kp.org; 4AJ Drexel Autism Institute, Drexel University, Philadelphia, PA 19104, USA; emkauffm@gmail.com; 5College of Health and Human Development, Penn State University, State College, PA 16801, USA

**Keywords:** Bayesian methods, mixtures, PBDEs, neurodevelopment

## Abstract

Objectives: Methods exist to study exposure mixtures, but each is distinct in the research question it aims to address. We propose a new approach focused on estimating both the summed effect and individual weights of one or multiple exposure mixtures: Bayesian Weighted Sums (BWS). Methods: We applied BWS to simulated and real datasets with correlated exposures. The analytic context in our real-world example is an estimation of the association between polybrominated diphenyl ether (PBDE) congeners (28, 47, 99, 100, and 153) and Autism Spectrum Disorder (ASD) diagnosis and Social Responsiveness Scores (SRS). Results: Simulations demonstrate that BWS performs reliably. In adjusted models using Early Autism Risk Longitudinal Investigation (EARLI) data, the odds of ASD for a 1-unit increase in the weighted sum of PBDEs were 1.41 (95% highest posterior density 0.82, 2.50) times the odds of ASD for the unexposed and the change in z-score standardized SRS per 1 unit increase in the weighted sum of PBDEs is 0.15 (95% highest posterior density −0.08, 0.38). Conclusions: BWS provides a means of estimating the summed effect and weights for individual components of a mixture. This approach is distinct from other exposure mixture tools. BWS may be more flexible than existing approaches and can be specified to allow multiple exposure groups based on a priori knowledge from epidemiology or toxicology.

## 1. Introduction

Environmental health research is increasingly oriented towards consideration of the health effects of multiple exposures, often referred to as complex mixtures [1,2]. This is natural, as humans are typically exposed to multiple chemicals simultaneously and those chemicals may act on similar biological pathways or may have similar effects on health. The understanding that environmental agents may act together is nothing new; it is embodied by concepts such as toxic equivalence, where one quantifies the relative contribution of exposures based on a reference exposure [3]. Aggregating a group of exposures into a single exposure using quantitative weights has been done to improve model parsimony, but faces multiple challenges: Weights do not directly translate to humans from animal models where they are commonly derived [4], estimating these weights from epidemiologic data has been difficult [5], and weights are rarely available for a specific outcome of interest [6]. Nonetheless, approaches that estimate the combined effect of a complex mixture and provide an understanding of the relative importance of each exposure to an outcome of interest are desirable.

To this end, methods have been advanced that allow for an understanding of both the overall effect of an exposure mixture as well as the relative importance of individual components of the mixtures. Notable among them are Bayesian Kernel Machine Regression (BKMR) and Weighted Quantile Sums (WQS). BKMR is able to model exposure mixtures in many ways, one of which provides the effect of an aggregate exposure mixture and the probability that each exposure is included in estimating that effect, similar to variable selection [7]. WQS provides a single, summed estimate of the effect of the mixture, which is restricted to be positive or negative, and targets so called ‘bad actors’ [8]. Both methods are informative for exposure mixtures research, but both are restricted in how they model the exposures of interest; they require that there is a single mixture, which does not allow for distinct mixture effects to be estimated for different groups of exposures. In addition, neither allows the integration of informative priors, which can be useful when prior evidence exists to inform effect estimation.

We propose a flexible Bayesian approach for studying complex exposure mixtures, which we will refer to as Bayesian Weighted Sums (BWS). The BWS can provide an estimate for a single, summed mixture effect as well as the percent contribution of individual components of the mixture to that effect. These quantities are distinct from those provided by BKMR and WQS, but are of substantial interest in exposure mixtures research. We implement this approach with a widely available Gibbs sampling software package. We demonstrate this approach with simulated data and apply it to a study of the effect of summed polybrominated diphenyl ethers (PBDEs) on neurodevelopment in the Early Autism Risk Longitudinal Investigation (EARLI) cohort. We then discuss the rationale for this approach and why this approach to studying mixtures may be appealing relative to others, including its ability to consider effects of exposures not subject to a summed mixture effect and flexibility in choice of exposure scales.

## 2. Materials and Methods

### 2.1. Rationale for Bayesian Weighted Sums

Exposures that occur as mixtures may be highly correlated, often because they arise from a single point source. When this is the case, identifying independent effects of each exposure may be undesirable, as none of the exposures can be reasonably expected to occur in isolation. This is often the case for classes of endocrine disruptors, such as polychlorinated biphenyls. However, the most common approaches to studying mixtures specify the model to explore independent or joint effects of individual exposures. Suppose we have measured five exposures; a regression model estimating the effects of those exposures on the outcome *Y* could be the following:(1)$$E\left[g\left(Y\right)\right]=\beta_{0}+\beta_{1}X_{1}+\beta_{2}X_{2}+\beta_{3}X_{3}+\beta_{4}X_{4}+\beta_{5}X_{5}$$

Here, *g* represents the link function for the regression model and *β_i_* represents the estimated effect of each exposure, *X_i_*. Researchers can choose from a variety of approaches, including penalized estimators, to improve estimation and identify exposure effects that are more strongly related to the outcome [9]. Alternatively, because the exposures often occur in tandem and an overall effect of the exposures may be of interest, we propose a model in which each exposure is weighted and the effect of the weighted summed exposures is estimated. This model is of the form:(2)$$E\left[g\left(Y\right)\right]=\theta_{0}+\theta_{1}\left(w_{1}X_{1}+w_{2}X_{2}+w_{3}X_{3}+w_{4}X_{4}+w_{5}X_{5}\right)$$

Here, θ_1_ represents the estimated effect of the weighted sum of exposures *X*_1_–*X*_5_, and *w*_1_–*w*_5_ are the estimated weights for each exposure of interest and are constrained to sum to 1. This is a straightforward reparameterization of model (1), but directly addresses a different research question [9]. That is, rather than estimating independent effects (*β_i_* in expression (1)), we estimate a single summed mixture effect (θ_1_). The weights represent a percent contribution of each exposure to that mixture effect. This approach is a natural extension of toxic equivalency factors estimated in toxicology [5]; but rather than using a single exposure as a reference, weights are estimated for each exposure in the model.

To estimate parameters of model (2), we adopt a Bayesian approach and apply a Dirichlet distribution to the weights. The Dirichlet distribution is a multivariate generalization of the beta distribution that has two desirable features for this problem [10]. First, the Dirichlet distribution automatically constrains the sum of all weights, *w*_1_…*w_k_*, to 1. Second, the values for the weights must be positive real numbers, thus ensuring that no exposures receive negative weights. We note here that while the weights are constrained to have positive values, the summed effect estimate, θ_1_, has no such constraints, and can take any value. The Dirichlet prior is specified so that: $w_{1}\ldots w_{5}\sim Dirichlet\left(\alpha_{1},\ldots ,\alpha_{5}\right),\;$where $\alpha_{k}=1.\;$This weak prior specification implies that, prior to observing any data, we think it most likely that all weights are equal. To complete the Bayesian specification of the model, we assume θ_1_ ~ N(μ = 0, σ^2^ = 100), a very weakly informative prior on these values. We note that one could use an informative prior on any of these effects, if prior information were available to inform that distribution. We provide the likelihood function and approximate posterior distribution as an Appendix A.

### 2.2. Simulations

Our simulations focus on various plausible scenarios in which a researcher might wish to estimate both the summed effect of multiple exposures and the individual percent contribution (i.e., weight) of each exposure to the summed effect. We simulated a single, continuous outcome and five exposures. We simulated datasets with 250, 500, and 1000 observations. Simulations at each sample size were repeated 500 times. We considered scenarios for which exposures were low to moderately correlated (corr = 0.1 to 0.5) and highly correlated (corr = 0.9), and for which the magnitudes of the summed effect were large ($\theta_{1}$ = 1.0) and small ($\theta_{1}$ = 0.2). In addition to simulations reported in the results section, we ran two additional simulations to gauge the impact of specifying negative weights, which violates the Dirichlet prior structure, and the impact of specifying a null summed mixture effect. We do not report these results, but discuss them as a limitation of our approach. Finally, simulations were also run where the standard error of the model was increased.

We conducted all simulations in the R statistical software package (V3.4.3) and used the Just Another Gibbs Sampler (JAGS) program. All models ran for 30,000 iterations with a burn-in of 3000 iterations and thinning of 2. Autocorrelation and trace plots were assessed for a subset of simulations to visually assess convergence. Gelman Rubin diagnostics were based on three chains and averaged across the 500 simulations to ensure that, on average, models converged. We also calculated mean squared error, average standard deviation (i.e., the mean of the estimated standard deviations), and 95% confidence interval coverage for the summed effect and weights for all simulations. We report median values for all estimated summed effects and weights.

### 2.3. Association of Summed PBDEs with ASD and SRS in the EARLI Cohort

We also applied our approach to estimate the association of prenatal exposure to summed polybrominated diphenyl ethers (PBDEs) with Social Responsiveness Scale (SRS) scores and diagnosis of Autism Spectrum Disorder (ASD) in the Early Autism Risk Longitudinal Investigation (EARLI) cohort study. The EARLI cohort is enriched for ASD risk, meaning participant mothers all had a previous child with an ASD diagnosis and were newly pregnant with a subsequent child (sibling). EARLI families were recruited at four sites (Drexel/Children’s Hospital of Philadelphia; Johns Hopkins/Kennedy Krieger Institute; UC Davis; and Kaiser Permanente of Northern California) from 2009–2012. In addition to having a biological child with ASD confirmed by EARLI study clinicians, to be eligible mothers also had to communicate in English or Spanish, be 18 years or older, live within 2 h of a study site, and be less than 29 weeks pregnant.

Evaluation of ASD in the younger sibling was conducted at the three-year old study visit. Classification of ASD is based on the Baby Siblings Research Consortium (BSRC) criteria [11]. This requires first meeting the Autism Diagnostic Observation Schedule (ADOS) criteria for ASD, and then the Diagnostic and Statistical Manual of Mental Disorders (version 5) criteria for ASD; the ADOS was administered by a trained assessor and the DSM criteria evaluation was done by a study clinician. Children not meeting BSRC criteria for ASD could be classified as having non-typical development based on their scores on the ADOS and the Mullen Scales of Early Learning. In addition, a dimensional measure of social responsiveness, a key trait underlying ASD, was assessed using the Social Responsiveness Scale (SRS), a 65-item parent-completed questionnaire. SRS has well-established psychometric properties with high validity, reliability, and reproducibility [12,13]. For primary analyses, typical and non-typical development children are grouped together as non-ASD cases. In addition, we z-score standardized the total SRS score, which accounts for child sex. We note here that analyses were conducted using z-score standardized SRS scores; these were nearly identical to t-score SRS analyses, the more standard metric, and, thus, are not reported.

We considered five PBDE congeners that were measured in EARLI: 28, 47, 99, 100, and 153. Lipid adjusted (ng/g) measures of PBDE biomarkers were obtained from serum samples collected from mothers during pregnancy; laboratory methods for measuring of PBDEs is described elsewhere [14]. We imputed values below the analytic limit of detection (LOD) by assuming those values to be equal to LOD/√2. Finally, we natural log transformed all PBDE values for primary analyses after correction for values < LOD.

We estimated crude and adjusted associations, the latter were adjusted for potential confounders identified before analyses: Maternal age (continuous), family income (ordinal, continuous), maternal race/ethnicity (non-Hispanic white (reference), Hispanic white, Black, Asian, other), study site (Drexel/Children’s Hospital of Philadelphia reference), female (reference) versus male sex at birth, and gestational age (continuous). All analyses used R and JAGS and ran for 80,000 iterations with a burn-in of 4000 and thinning of 2. Trace and autocorrelation plots were examined for indication of model convergence. Gelman-Rubin diagnostics were based on three chains, and values of 1.0 indicated that parameter estimates from each individual chain resulted in the same posterior distribution. We visually examined trace and autocorrelation plots to ensure models did not fail to converge. We discuss median values as the effect estimates of interest. However, because the distributions of exposure weights are not normal, we provide mean values to illustrate that, on average, exposure weights sum to 1. We also separately fit frequentist logistic regression models that (1) estimated each PBDE effect individually and (2) adjusted for all PBDE simultaneously. We estimated the independent effects for each PBDE from these models.

All code to recreate simulation results and EARLI study results will be available via an eSupplement (simulations) as well as via the corresponding author’s professional webpage (http://ghassanbhamra-phd.org). Data to recreate results from the EARLI cohort study requires permission from the study principal investigators.

## 3. Results

### 3.1. Simulations

Table 1 summarizes the simulation results of a summed mixture effect and the weights of five exposures that are components of the mixture. We constructed 24 scenarios with varying correlation structures, sample sizes, and magnitudes of summed effects. We only report 8 scenarios because for 16, in which we increased the standard error and sample size of the model, the results are similar and, thus, not additionally informative. The estimate of the summed effect, *θ*_1_, is approximately unbiased. In addition, these estimates have relatively low MSE and bias, and expected or near expected 95% confidence interval coverage in all scenarios. As would be expected, MSE and average bias decreases as sample size increases, and as the standard error is reduced. Notably, MSE, average bias, and average standard deviation of *θ*_1_ decreased when the five exposures were highly correlated vs. low to moderately correlated. For example, with a sample size of 250, *θ*_1_ = 1.0 and standard deviation of the linear model = 0.5, the MSE, average bias, and average standard deviation fell from 0.003 to 0.001, −0.013 to 0.001, and 0.053 to 0.032; this decrease in MSE and average bias for the overall effect was offset by a corresponding increase in these quantities for the weights as the correlation increased. Simulations for a negative summed effect were conducted and had similar performance (data not shown).

Similar to the estimated summed mixture effect, the MSE, average bias, and average standard deviation of the weights show steady improvement as sample size increases and standard error of the outcome decreases. Unlike the summed mixture effect, MSE and average standard deviation of the weights increase as the correlation of the exposures is changed from low or moderate to high for all five exposures. This is true across all scenarios. Unlike the summed effect, 95% confidence interval coverage tended to exceed nominal levels. However, we do note some tendency toward expected coverage as sample sizes increase and when correlations across the five exposures are lower.

### 3.2. EARLI Results

Table 2 describes the distributions of demographic characteristics for ASD cases and non-cases. Maternal age at enrollment and gestational age of the children were similar between cases and non-cases. As expected, children with ASD were disproportionally male and had higher SRS scores than non-cases. Further, a higher proportion of mothers of ASD cases represented racial/ethnic minorities compared to non-cases. Figure 1 illustrates the correlation structure of the PBDEs we consider. Notable is that four of five exposures are highly correlated, with r^2^ > 0.75; only PBDE 153 is not moderately to highly correlated with the remaining four PBDEs (28, 47, 99, and 100).

Crude and adjusted models show positive but imprecise associations between summed PBDE exposures with increasing SRS scores and ASD risk. Results are summarized in Table 3. Trace plots indicated excellent model performance (Appendix A). The change in SRS z-scores (and corresponding 95% HPD) per 1-unit increase in the weighted sum of PBDE exposures are 0.25 (0.05, 0.45) and 0.15 (−0.08, 0.38) for crude and adjusted models, respectively. Notably, the 95% HPD intervals mostly overlap and are of similar width at 0.40 (crude) and 0.46 (adjusted). The odds ratio of ASD per 1-unit increase in weighted sum of PBDE exposures are 1.29 (0.83, 1.99) and 1.41 (0.82, 2.50) for crude and adjusted models, respectively. Confidence limit widths (upper minus lower bound) were more precise for crude models compared to adjusted models at 1.16 and 1.68, respectively. Similar to SRS, the 95% HPD intervals overlap a great deal. In all four models, the PBDE weights show similar patterns. For SRS crude and adjusted models, PBDEs 28 and 47 explain slightly more of the summed effect than PBDEs 99 and 100, and all four explain more of the effect than PBDE 153. For ASD crude and adjusted models, PBDEs 28, 47, and 100 explain slightly more of the summed effect than PBDEs 99 and 100.

Table 4 summarizes results from single exposure models for PBDEs and models that estimated independent effects of all PBDEs within a single model. Notably, the model for the association of PBDEs and SRS that included all PBDEs that failed to converge and, thus, quantitative effect estimates are not provided. In single exposure models, effect estimates for PBDEs are generally consistent with the overall effect estimate from the Bayesian weighted sums model. The notable exception is PBDE 153, for which associations with both SRS and ASD are closer to a null effect; however, the estimated weights for PBDE 153 in the BWS model were generally the smallest of all the weights. When estimating the association of all PBDEs to ASD in a single model, effect estimates are notably imprecise and suggest associations in both positive and negative directions; this is inconsistent with both the Bayesian weighted sums model and models where PBDEs effects are estimated independent of one another.

## 4. Discussion

We illustrated our Bayesian Weighted Sums (BWS) approach for estimating both the summed effect of a complex mixture as well as the percent of that effect explained by components of the mixture. Simulation results demonstrated that the approach performs well across a variety of plausible scenarios. We applied this approach to estimate the association of a mixture of PBDEs with both SRS scores and ASD risk in a high-risk longitudinal cohort study. Results support a positive but imprecise association between summed PBDE exposures and both outcomes in crude and adjusted models. Further, there was variation in the percentage of the summed effect explained by each exposure. Results are largely consistent with simple frequentist models that estimate independent exposure effects excluding co-exposures; however, frequentist models that adjusted for all PBDEs simultaneously produced model effects with very large standard errors or simply failed to converge. Below, we discuss our biological and practical motivations for BWS, benefits that distinguish it from similar approaches, and potential limitations.

BWS is desirable for studying exposure mixtures because components are often believed to act on similar biological pathways; this may be due to structural similarities in compounds or toxicological evidence supporting this belief. Evidence from toxicology has led to the development of weighting techniques for combining exposures; specifically, toxic equivalency and relative potency factors aim to combine multiple exposures by weighting them based on their theorized toxicity relative to a reference exposure [15]. Although designed with risk/hazard assessment in mind, these toxic equivalence factors are appealing to researchers interested in reducing the dimensionality of complex mixtures. Using toxic equivalency is more defensible than assuming equal toxicity and summing exposures (ex: All PBDEs). While appealing, these quantities do not directly translate from animal or cellular experimental research to observational human research [6]. Further, we have previously demonstrated that these weighting factors that correspond to toxic equivalency factors cannot be estimated from observational epidemiology data [5]. Finally, these quantities treat toxicity of components as equivalent to that of a reference compound assuming the weights applied are accurate. For example, individual dioxin compounds are down-weighted and then summed based on their toxicity relative to 2, 3, 7, 8 TCDD, the most toxic dioxin compound [16]. This conversion reduces the question of the health effects of complex exposure mixtures down to a question of the health effects of the reference exposure, as we are unable to quantify the effects of specific dioxins after they are weighted and summed. BWS draws from the concept of toxic equivalence to provide a tractable alternative for estimating effects of mixture components and their summed effects without making assumptions about which compounds are the most or least toxic.

Practically, it may not make sense to study individual components of a complex mixture because they do not occur in isolation. Our application of BWS to study PBDEs offers a case in point. Four of the five PBDEs we examine are highly correlated, PBDE 153 being the only exception. The highly correlated nature of PBDEs has been demonstrated elsewhere [17]; this suggests that any public health intervention designed for a single PBDE necessarily changes exposure to all of them. Nonetheless, it may also be of interest to understand whether PBDE congeners, or components of any complex exposure mixture, seem to explain more or less of the health effect of the mixture. BWS is a practical solution to this challenge.

It is important to note that we cannot directly compare BWS to other, recently developed statistical approaches to study exposure mixtures. Some approaches appear similar to BWS, but are fundamentally distinct in the research question they aim to address. A two-step optimization based approach was suggested previously that targets a single mixture effect and a percent contribution of the individual components, though we have not seen this approach practiced [18]. BWS may be seen as similar to three contemporary approaches designed to measure the effect of a mixture: Bayesian Kernel Machine Regression (BKMR), Weighted Quantile Sums (WQS), and quantile g-computation. While these methods provide a single mixture effect, each provides a fundamentally different quantity based partly on the way it handles the exposure matrix and, thus, they cannot be directly compared via simulation. Nonetheless, we can qualitatively compare these approaches to inform potential users of when each might be best applied. BKMR is capable of estimating a summary mixture effect as well as providing estimates of the posterior inclusion probability for individual mixture components (which can be zero, meaning those components are dropped from the model) [7]. Thus, certain components may be entirely excluded from contributing to the mixture effect with BKMR. In contrast, all mixture components can be included in the BWS framework, though this may be manipulated if the Dirichlet prior is changed to a variable selection oriented prior. WQS is designed to provide a summed mixture effect and the percent contribution to that effect of each mixture component. However, WQS is designed to highlight so called ‘bad actors,’ depending on the direction the effect is restricted to; so, for a positive restricted effect, exposures that align with a positive effect are fundamentally biased away from their true values; this was illustrated in the paper presenting quantile g-computation, which is an approach oriented toward estimating effects of hypothetical public health interventions on mixtures [19]. Simulations demonstrated that BWS can identify positive or negative summed effects without prior restrictions. BWS is also capable of accommodating any exposure scale that is desired by the researcher. When mixture components demonstrate different distributions, a z-score may make sense. However, if the distributions are similar on a raw scale, such as μg/m^3^ for air pollutants, users can decide to maintain this exposure scale.

The flexibility of BWS allows for consideration of many complex mixture questions. Users need not restrict the complex mixture problem to a single summed effect. Rather, users might consider multiple summed effects for different exposure groupings within a single regression model. For example, one could estimate the summed effect of PBDEs and PCBs and the percent contribution of components within each group to their respective summed effect. This approach may be desirable because PBDEs and PCBs demonstrate high within chemical class correlations. Further, they are both believed to impact neurodevelopment via endocrine disruption: Both have been shown to disrupt thyroid hormones [20]. This supports the study of these chemicals within a single regression model, which can be achieved with BWS. We conducted all analyses with the JAGS software package. However, it is worth noting that other approaches that sample from the posterior distribution, such as STAN, could just as easily work for implementing our proposed method.

There are some notable limitations of our approach. First, when the true summed mixture effect is zero, different weights cannot be reliably estimated. If the summed mixture effect is zero, then there is no effect for the exposures to explain. Indeed, when we ran simulations to test this, the estimated weights were always approximately equal regardless of their simulated true values. This would represent a substantial problem if some exposures had a positive effect while others had a negative effect, cancelling each other out. Related, even if the exposures having different directions of effect did not exactly cancel out, estimation problems could still arise. For instance, there is a correspondence between the beta coefficients in expression (1) and the weights in expression (2). If beta coefficients in expression (1) are of different signs, it would imply that weights in expression (2) could be negative. Because the Dirichlet prior truncates the parameter space for the weights, negative weights are not possible, and estimation will be biased. Fortunately, these limitations are overcome by using prior knowledge and crude examination of the data to recode covariates such that all effects are in the same direction, if appropriate. In our example, we assume that no PBDE congener considered can protect against ASD risk or any individual domains of the SRS score. Because our model is flexible, mixture components thought to have opposing effects can be modeled separately or within different positive or negative effect groups. To our knowledge, this flexibility is not available with other existing mixtures approaches. Finally, BWS requires some basic knowledge of directly coding a Bayesian model. This is distinct from other methods outlined, which are available as plug-and-play packages in R software. However, we believe most epidemiologists will find the approach accessible, and we provide detailed, annotated code to both replicate our simulation results and apply BWS.

The study of complex exposure mixtures is a somewhat new and rapidly developing area of environmental health research. As such, many statistical tools exist and are under development to apply to this area. In fact, a critical review of the field outlined many of these methods [1]. We have noted similarities and differences between BWS and other, somewhat similar statistical methods, including BKMR. It is additionally worth noting that there are many additional tools that could be useful for studying exposure mixtures in the broad area of penalized estimators (LASSO, elastic net) and regression trees (random forest, Bayesian Additive Regression Trees). However, we want to note, as in the critical review by Hamra and Buckley [1], that the latter methods are ideally suited to higher dimensional exposure mixtures research, where there is a fundamental need to shrink parameter estimates or eliminate exposures from a model to gain useful insights into health effects of exposures. Our method, and the others described in more detail, focus more on lower dimensional problems.

## 5. Conclusions

In conclusion, BWS is a flexible approach to estimate the summed effects of complex mixtures and the percent contribution of components to that effect. Simulations demonstrated reliable performance of BWS. Application of BWS to study summed PBDEs with SRS scores and ASD risk showed positive associations for both outcomes with variation in the percent contribution of PBDEs, but somewhat consistent results for the percent contribution of individual congeners. We believe researchers concerned with complex exposure mixtures can benefit from the application of BWS to their research questions.

## Figures and Tables

**Figure 1 ijerph-18-01373-f001:**
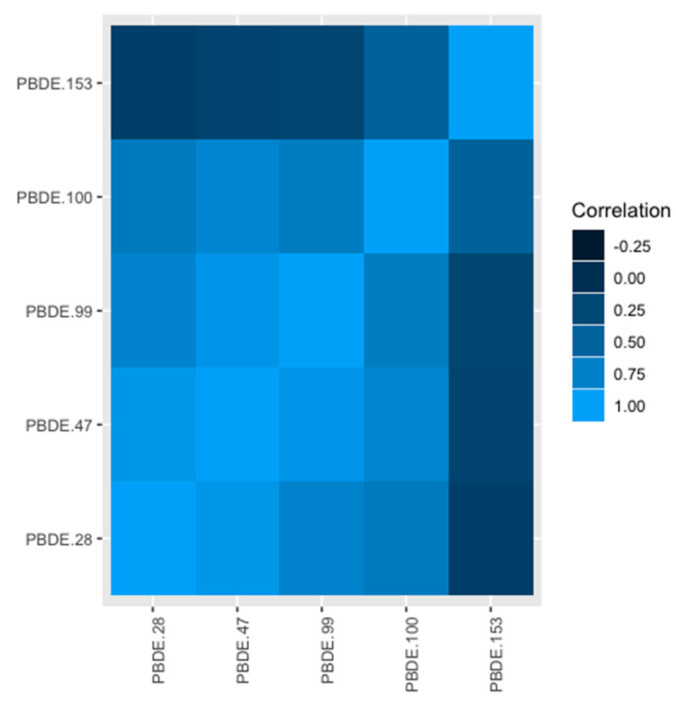
Correlations of polybrominated diphenyl ethers (PBDEs) measured in maternal biospecimens in the Early Autism Risk Longitudinal Investigation (EARLI) cohort.

**Table 1 ijerph-18-01373-t001:** Simulation results. Weights are specified so that *w*_1_ = 0.1, *w*_2_ = 0.3, *w*_3_ = 0.2, *w*_4_ = 0.1, *w*_5_ = 0.3.

	β = 1.0, Standard Error = 0.5											
	N = 250						N = 500					
Correlation	Coefficient	Estimate	Average Standard Error	Average Bias	MSE	95% CI Coverage		Estimate	Average Standard Error	Average Bias	MSE	95% CI Coverage
Low and moderate	*θ*	0.99	0.053	−0.013	0.003	95%	*θ*	1.00	0.035	−0.004	0.001	95%
	*w* _1_	0.10	0.036	−0.002	0.001	96%	*w* _1_	0.10	0.025	−0.001	0.001	95%
	*w* _2_	0.31	0.048	0.005	0.002	95%	*w* _2_	0.30	0.031	0.001	0.001	94%
	*w* _3_	0.20	0.045	−0.005	0.002	94%	*w* _3_	0.21	0.031	0.002	0.001	96%
	*w* _4_	0.10	0.043	0.005	0.002	97%	*w* _4_	0.10	0.031	−0.001	0.001	96%
	*w* _5_	0.30	0.037	−0.003	0.001	96%	*w* _5_	0.30	0.025	−0.002	0.001	95%
High	*Θ*	1.00	0.032	0.001	0.001	95%	*θ*	1.00	0.024	−0.001	0.001	95%
	*w* _1_	0.11	0.068	0.024	0.003	97%	*w* _1_	0.10	0.052	0.008	0.002	98%
	*w* _2_	0.28	0.084	−0.024	0.006	95%	*w* _2_	0.29	0.064	−0.006	0.004	96%
	*w* _3_	0.18	0.079	−0.008	0.005	97%	*w* _3_	0.19	0.061	−0.013	0.003	95%
	*w* _4_	0.11	0.071	0.028	0.003	98%	*w* _4_	0.10	0.053	0.013	0.002	97%
	*w* _5_	0.28	0.083	−0.021	0.006	95%	*w* _5_	0.30	0.061	−0.002	0.004	95%
	**β = 0.2 Standard Error = 0.1**											
	**N = 250**						**N = 500**					
**Correlations**	**Coefficient**	**Estimate**	**Average Standard Deviation**	**Average Bias**	**MSE**	**95% CI Coverage**		**Estimate**	**Average Standard Deviation**	**Average Bias**	**MSE**	**95% CI Coverage**
Low and moderate	*θ*	0.20	0.011	−0.003	0.000	95%	*θ*	0.20	0.007	−0.001	0.000	95%
	*w* _1_	0.10	0.036	−0.002	0.001	97%	*w* _1_	0.10	0.025	−0.001	0.001	95%
	*w* _2_	0.31	0.049	0.005	0.002	95%	*w* _2_	0.30	0.031	0.001	0.001	94%
	*w* _3_	0.20	0.046	−0.005	0.002	94%	*w* _3_	0.21	0.031	0.002	0.001	96%
	*w* _4_	0.10	0.043	0.005	0.002	96%	*w* _4_	0.10	0.031	−0.001	0.001	96%
	*w* _5_	0.30	0.038	−0.003	0.001	96%	*w* _5_	0.30	0.025	−0.002	0.001	95%
High	*θ*	0.20	0.006	0.000	0.000	95%	*θ*	0.20	0.005	0.000	0.000	95%
	*w* _1_	0.11	0.068	0.025	0.003	97%	*w* _ 1 _	0.10	0.052	0.008	0.002	98%
	*w* _2_	0.28	0.084	−0.024	0.006	96%	*w* _2_	0.29	0.064	−0.006	0.004	96%
	*w* _3_	0.18	0.079	−0.008	0.005	97%	*w* _3_	0.19	0.061	−0.013	0.003	95%
	*w* _4_	0.11	0.071	0.028	0.003	98%	*w* _4_	0.10	0.053	0.013	0.002	97%
	*w* _5_	0.28	0.084	−0.021	0.006	95%	*w* _5_	0.30	0.061	−0.002	0.004	94%

**Table 2 ijerph-18-01373-t002:** Demographic description of EARLI cohort participants.

	Autism Spectrum Disorder (ASD) (N = 42)	No ASD (N = 124)
Maternal Age	33 (5.4)	34 (4.9)
Gestational age	38 (2.8)	39 (2.0)
Social Responsiveness Scores (SRS) t-score	62 (14)	49 (8)
Site		
Drexel	6 (14%)	34 (27%)
Johns Hopkins	9 (21%)	32 (26%)
Kaiser Permanente	16 (38%)	33 (27%)
UC Davis	11 (26%)	25 (20%)
Family income		
<$10,000	4 (9.5%)	4 (3.2%)
$10,000 to <$20,000	0 (0.0%)	3 (2.4%)
$20,000 to <$30,000	3 (7.1%)	4 (3.2%)
$30,000 to <$50,000	2 (4.8%)	22 (18%)
$50,000 to <$75,000	8 (19%)	17 (14%)
$75,000 to <$100,000	8 (19%)	19 (15%)
$100,000 to <$200,000	11 (26%)	40 (32%)
$200,000 or more	0 (0.0%)	12 (9.7%)
missing	6 (14%)	3 (2.4%)
Maternal Race-Ethnicity		
White, non-Hispanic	20 (48%)	72 (58%)
White, Hispanic	4 (9.5%)	9 (7.3%)
Black	6 (14%)	12 (9.7%)
Asian	5 (12%)	16 (13%)
Other	7 (17%)	15 (12%)
Male birth sex	34 (81%)	59 (48%)

**Table 3 ijerph-18-01373-t003:** Summed effect of a PBDE mixture on SRS scores and ASD in the EARLI cohort.

Scheme				ASD Crude (Top) and Adjusted (Bottom)			
	Median	Mean	95% HPD		Median	Mean	95% HPD
*θ*	0.25	0.25	(0.05, 0.45)	*θ*	1.28	1.29	(0.83, 1.99)
w(PBDE28)	0.19	0.22	(0.00, 0.56)	w(PBDE28)	0.21	0.25	(0.00, 0.61)
w(PBDE47)	0.21	0.24	(0.00, 0.59)	w(PBDE47)	0.15	0.19	(0.00, 0.51)
w(PBDE99)	0.17	0.21	(0.00, 0.54)	w(PBDE99)	0.15	0.19	(0.00, 0.50)
w(PBDE100)	0.15	0.19	(0.00, 0.52)	w(PBDE100)	0.17	0.21	(0.00, 0.55)
w(PBDE153)	0.10	0.13	(0.00, 0.34)	w(PBDE153)	0.12	0.16	(0.00, 0.45)
	**Median**	**Mean**	**95% HPD**		**Median**	**Mean**	**95% HPD**
*θ*	0.15	0.15	(−0.08, 0.38)	*θ*	1.41	1.41	(0.82, 2.50)
w(PBDE28)	0.19	0.23	(0.00, 0.57)	w(PBDE28)	0.18	0.22	(0.00, 0.56)
w(PBDE47)	0.19	0.23	(0.00, 0.58)	w(PBDE47)	0.17	0.21	(0.00, 0.55)
w(PBDE99)	0.15	0.19	(0.00, 0.50)	w(PBDE99)	0.14	0.18	(0.00, 0.48)
w(PBDE100)	0.15	0.19	(0.00, 0.51)	w(PBDE100)	0.18	0.22	(0.00, 0.57)
w(PBDE153)	0.13	0.16	(0.00, 0.44)	w(PBDE153)	0.13	0.17	(0.00, 0.46)

**Table 4 ijerph-18-01373-t004:** Independent and summed effects of PBDEs estimated with traditional regression techniques. Summed effects are per 1 unit increase in natural logged exposure values.

	Models Include Each Exposure Individually				Single Model Includes All Exposures	
	SRS		ASD		ASD	
	Mean Difference	95% CI	Odds Ratio	95% CI	Odds Ratio	95% CI
PBDE28	0.20	(−0.05, 0.45)	1.53	(0.81, 2.96)	1.09	(0.25, 4.70)
PBDE47	0.16	(−0.01, 0.33)	1.38	(0.89, 2.17)	3.19	(0.53, 24.32)
PBDE99	0.12	(−0.04, 0.28)	1.14	(0.76, 1.71)	0.25	(0.06, 0.91)
PBDE100	0.13	(−0.05, 0.31)	1.42	(0.89, 2.28)	2.20	(0.61, 8.11)
PBDE153	0.01	(−0.15, 0.17)	0.93	(0.59, 1.44)	0.75	(0.38, 1.42)
summed PBDEs	0.10	(−0.10, 0.30)	1.30	(0.78, 2.18)	n/a	n/a

## Data Availability

The code necessary to recreate simulated data and results are available from the cooresponding author. EARLI data may be available upon request.

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
