# Peer review of "Bayesian Weighted Sums: A Flexible Approach to Estimate Summed Mixture Effects"

_ijerph, 2021, doi:10.3390/ijerph18041373_

Round 1

Reviewer 1 Report

This manuscript proposes a method called Bayesian Weighted Sums in order to estimate in a joint way the mixture effect and the individual weights. In order to estimate the parameters of interest, the authors adopt a Bayesian approach with prior distributions given by normal and Dirichlet distributions. The estimation procedure was carried out in the JAGS program. In my opinion, from the statistical or data analysis point of view, no new computational
or data-analytic method was discussed in the paper. Section 2, which is the main part of the paper in my opinion, does not inform the reader that the estimation procedure is based on the joint posterior distribution. The authors are encouraged to include a discussion about how to get the Likelihood function and the posterior distribution for the parameters of interest. What
is the main reason for using the JAGS program? In addition, as the paper is described, it is very difficult to evaluate the innovation and performance of the proposed method in relation to existing tools. The authors are encouraged
to develop a systematic comparison between the proposed method and some state-of-the-art available using simulated datasets.
I have some other comments:
(1) The authors need to explicitly articulate what is the main difference between their proposed manuscript and the published ones. The comparison criterion should be cited in the Introduction. The method used for estimation of the paramaters also should be cited in the Introduction.
(2) pag.2, section 2. In my opinion, this section is the main part of the paper. However, it is very confusing. The authors are encouraged to present the likelihood function and the posterior distribution for the parameters of the model presented.
(3) pag2. expressions (1) and (2). Both model are kinds of regression models?
• If yes, should not there exists a random error term in the modeling procedure? How to estimate the variance of the random error under the Bayesian approach described?
• If no, please give a justification.
(4) pag.3, line 95. “To estimate model (2)” or “To estimate parameters of the model (2)?”
(5) pag.3, line 114. Why θ1 = 1, 0 is considered large? An user could set θ1 = exp(4.61)? If yes, the author are encouraged to include a discussion on this case  1
(6) pag.3, line114-117. “In addition to simulations reported in the results section, we ran two
additional simulations to gauge the impact of specifying negative weights, which violates
the Dirichlet prior structure, and the impact of specifying a null summed mixture effect”.
Why the interest in this case if it violates the prior structure? For this case, another
modeling procedure should be considered?
(7) pag.3, line 120. The author informe the reader on the use of JAGS. At this point, some justification should be given. If I am not wrong, the JAGS program uses a slice sampler algorithm in order to get samples from the posterior distribution. How difficult is implement a Metropolis-Hastings algorithm for this problem? Why a user should opt to use the JAGS instead to implement a Metropolis-Hastings algorithm?
(8) The authors are encouraged to make a systematic comparison between proposed method and some state-of-the-art available using the simulated datasets. As it stands, it is very difficult to evaluate the innovation and performance on the proposed method in relation to existing tools.
(9) The authors could devote few lines to justify the convergence of the proposed method

Author Response

R1 writes: In my opinion, from the statistical or data analysis point of view, no new computational or data-analytic method was discussed in the paper. Section 2, which is the main part of the paper in my opinion, does not inform the reader that the estimation procedure is based on the joint posterior distribution. The authors are encouraged to include a discussion about how to get the Likelihood function and the posterior distribution for the parameters of interest. What is the main reason for using the JAGS program?

Response: In fact, we propose a Bayesian application of a previously recommended approach to estimate the summed effect of a mixture and individual weights of the mixture components. We state this on line 404 of the paper. Like many other publications have demonstrated, we believe that part of the appeal is that estimating exposure weights is easier and far more flexible when applying a Bayesian approach. The prior application, using likelihood-based methods, has not been widely applied. We believe that one reason is the lack of a clear and accessible implementation approach. Here, we provide a Bayesian technique which is easily applied. We use the JAGS program because it is very widely used and freely available; however, this technique could just as easily be applied with other sampling procedures, such as STAN. We have added text to note this in the manuscript. (Line 442)

R1 writes: In addition, as the paper is described, it is very difficult to evaluate the innovation and performance of the proposed method in relation to existing tools. The authors are encouraged
to develop a systematic comparison between the proposed method and some state-of-the-art available using simulated datasets.

Response: It is not possible, unfortunately, to directly compare the performance of our approach to other recent methods, such as weighted quantile sums and Bayesian Kernel Machine Regression. This is a notable problem in the mixtures literature: each method is fundamentally distinct in that the estimand of interest is completely different. Thus, there can be no direct comparison across methods. As an alternative, we provide detailed simulations to show our approach performs as expected, in the case where we know the true summed mixture effect and exposure weights. We have emphasized this in a paragraph of the discussion outlining the differences between BWS and other approaches, and we have added text to emphasize this point. (Line 402)

R1 writes: The authors need to explicitly articulate what is the main difference between their proposed manuscript and the published ones. The comparison criterion should be cited in the Introduction. The method used for estimation of the paramaters also should be cited in the Introduction.

Response: We have added text to the introduction to distinguish our method and note use of MCMC sampling for estimation. We also note that use of these Bayesian methods increases the flexibility, allowing researchers to incorporate substantive knowledge about possible weights or overall effects. (Lines 131,391)

R1 writes: (2) pag.2, section 2. In my opinion, this section is the main part of the paper. However, it is very confusing. The authors are encouraged to present the likelihood function and the posterior distribution for the parameters of the model presented.

Response: We appreciate the reviewer’s desire for more information on this point. However, in an applied journal such as IJERPH, we feel strongly that writing out the likelihood will not add to the understanding of the journal’s readers. Further, the posterior is not available in closed form, hence the need for MCMC methods. We believe that explaining the model in terms of regressions, which epidemiologists are familiar with, is the clearest approach.

R1 writes: pag2. expressions (1) and (2). Both model are kinds of regression models?
• If yes, should not there exists a random error term in the modeling procedure? How to estimate the variance of the random error under the Bayesian approach described?
• If no, please give a justification.

Response: We appreciate the opportunity to clarify. We have specified the regressions in a generic form to accommodate all general linear models. We have clarified that the function in expressions 1 and 2 are the expected values of the transformed linear predictor. The regression itself may or may not have an error term. For instance, with a linear regression and error would be included but for a logistic model it would not. In either approach, the expected value would not have an error term, so remains general. We have clarified that the expectation function for expressions 1 and 2 (Line 152)

R1 writes: pag.3, line 95. “To estimate model (2)” or “To estimate parameters of the model (2)?”

Response: We have edited this text as advised. (Line 132)

R1 writes:  pag.3, line114-117. “In addition to simulations reported in the results section, we ran two additional simulations to gauge the impact of specifying negative weights, which violates
the Dirichlet prior structure, and the impact of specifying a null summed mixture effect”.
Why the interest in this case if it violates the prior structure? For this case, another
modeling procedure should be considered?

Response: This simulation is used to evaluate the performance of our model if the user misspecifies the prior distribution. This is a natural concern as researchers may have little knowledge about these parameters. These simulations demonstrate the robustness of the method.

R1 writes: (7) pag.3, line 120. The author informe the reader on the use of JAGS. At this point, some justification should be given. If I am not wrong, the JAGS program uses a slice sampler algorithm in order to get samples from the posterior distribution. How difficult is implement a Metropolis-Hastings algorithm for this problem? Why a user should opt to use the JAGS instead to implement a Metropolis-Hastings algorithm?

Response: There are many ways of sampling from the posterior distribution and all of them should theoretically arrive at the same answer. Slice sampling, Gibbs, and M-H are all simply algorithms to arrive at an answer and user choice will primarily be dictated by ease of use in this field. As we mentioned above we chose JAGS because it is very commonly used and free, which will allow readers of this journal to use the method if they so desire. We have also made our code publicly available for this purpose. (Line 401)

R1 writes: The authors could devote few lines to justify the convergence of the proposed method

Response: We appreciate the comment. We have included trace plots in supplementary material and now state that convergence was assessed via visual inspection of trace plots and Gelman Rubin diagnostics. (Line 216)

Reviewer 2 Report

This paper proposes a new approach to estimate both the summed effect and individual weights of a group of exposures: Bayesian Weighted Sums (BWS).
The work presented in the article is interesting, but a few suggestions need to be implemented before acceptance.
1. Please describe the dataset and the evaluation criteria used.
2. The introduction is too short. The authors do not explain how/why the proposed method address the problems.
3. It seems that there are more than one problem statements which the authors have tried to address. Accordingly, there should be multiple contributions
4. The related work must cite recent and more related work from the literature.
5. The critical analysis of the literature must be provided.
6. The referencing styles must be consistent and improved.
7. In line 71, please cite the reference.

Author Response

R2 writes: Please describe the dataset and the evaluation criteria used.

Response: We are uncertain what reviewer 2 is asking for here. However, we would like to note that the simulation data are described and statistical code to recreate all simulations are available. Also, we briefly describe the EARLI cohort data and cite relevant articles providing further details about the cohort.

R2 writes: The introduction is too short. The authors do not explain how/why the proposed method address the problems.

Response: We have explained the utility of BWS in the discussion section rather than the introduction, where we also distinguish it from other methods that are applied to study exposure mixtures. We believe that this is the most appropriate section of the manuscript for such a discussion.

R2 writes: It seems that there are more than one problem statements which the authors have tried to address. Accordingly, there should be multiple contributions

Response: We are unclear what the reviewer means by this. If they could clarify, we would be happy to address this comment.

R2 writes: The related work must cite recent and more related work from the literature… The critical analysis of the literature must be provided.

Response: To our knowledge, we have cited the most contemporary methods that have been applied to lower dimension exposure mixtures problems, like what we present here. This includes what we understand the be the most recently developed quantile g computation (2020). We should clarify that other methods, such as LASSO, elastic net, random forest, or similar, are not comparable to what we present here as they focus largely on data reduction be elimination or shrinkage of higher dimensional exposures. We have added a paragraph to note this and to address the reviewers desire for a critical analysis of the literature. (Line 429)

R2 writes: The referencing styles must be consistent and improved.

Response: the citation format has been corrected.

Reviewer 3 Report

Of the five PBDEs, one(153)  is not correlated to the others.

What is its effect in your summed mixture estimation ?

Since the distributions of exposures are not normal one cannot 

really speak of standard deviations and everything reduces to reporting  

median values.  

Author Response

R3 writes: Of the five PBDEs, one(153)  is not correlated to the others. What is its effect in your summed mixture estimation?

Response: We ran the models excluding PBDE 153. Exclusion did not materially change the summed mixture effect estimate and, thus, we did not further discuss this finding in the manuscript.

R3 writes: Since the distributions of exposures are not normal one cannot really speak of standard deviations and everything reduces to reporting median values.  

Response: We would like to clarify that our discussion focuses on the estimated parameters, rather than the exposure distributions. We note here that our simulated exposures are, in fact, normally distributed. Regarding our application to EARLI cohort data, we take the log of exposures which are, as the reviewer notes, not normally distributed. We do not otherwise discuss the distribution of the exposures or report their standard deviation.

Reviewer 4 Report

The paper discusses an important subject. The advances in the data-acquiring process and the large extend of systems necessitates having efficient tools, such as the one proposed in this draft. The paper is well written and organized. My comments are as follows:

While the abstract has aimed to provide a comprehensive overview of the main contribution, there is a need to be revised so that the general reader can grasp the main idea/topic of the draft and the main contribution. 

-Although a good discussion about the proposed framework's superiority is provided in terms of the numerical results, discussion about the proposed framework's complexity and how it compares with the existing techniques are highly recommended. 

-Having a nice schematic diagram in the draft would be helpful. This alleviates the difficulty of going to details of the techniques for the readers.

- There has been a surge in the application of Machine Learning and Statistical framework to solve similar problems focused in this paper. The authors are encouraged to include some of the recent articles in the introduction to give an excellent holistic overview of the existing techniques to general readers:

# "Bayesian estimation of multicomponent relaxation parameters in magnetic resonance fingerprinting." Magnetic resonance in medicine 80.1 (2018): 159-170.

# "Scalable optimal Bayesian classification of single-cell trajectories under regulatory model uncertainty." BMC genomics 20.6 (2019): 1-11.

# "Boolean Kalman filter and smoother under model uncertainty." Automatica 111 (2020): 108609

- The format of some of the references is not in standard form. These need to be checked and fixed.

Author Response

R4 writes: While the abstract has aimed to provide a comprehensive overview of the main contribution, there is a need to be revised so that the general reader can grasp the main idea/topic of the draft and the main contribution. (Line 25)

Response: We have edited the abstract as advised to focus on the main idea of the manuscript.

R4 writes: Although a good discussion about the proposed framework's superiority is provided in terms of the numerical results, discussion about the proposed framework's complexity and how it compares with the existing techniques are highly recommended. 

Response: We have added text to note that BWS requires some coding knowledge, but that we provide all necessary code to apply it. (Line 424)

R4 writes: There has been a surge in the application of Machine Learning and Statistical framework to solve similar problems focused in this paper. The authors are encouraged to include some of the recent articles in the introduction to give an excellent holistic overview of the existing techniques to general readers:

# "Bayesian estimation of multicomponent relaxation parameters in magnetic resonance fingerprinting." Magnetic resonance in medicine 80.1 (2018): 159-170.

# "Scalable optimal Bayesian classification of single-cell trajectories under regulatory model uncertainty." BMC genomics 20.6 (2019): 1-11.

# "Boolean Kalman filter and smoother under model uncertainty." Automatica 111 (2020): 108609

Response: The suggested citations are interesting, but more oriented towards higher dimensional data reduction challenges. Our proposed method is more suited to lower dimension exposure problems. We note this with a new paragraph discussing BWS to other existing methods. (Line 429).

Round 2

Reviewer 1 Report

Please, find attached my comments.

Author Response

We appreciate the reviewer's continued interest in our work! We provide the likelihood function and approximate posterior distribution as an eAppendix for more technically savy readers. (line 138)

We also note here that there is no label switching problem with our specification of the prior, which is a dirichlet prior that constrains all values to be between 0 and 1.

Reviewer 2 Report

Authors have incorporated my suggestions and improved the paper. I think the paper is in acceptable format.

Author Response

We appreciate the reviewer's positive feedback, and comments.

Reviewer 4 Report

The paper is well-revised. The references, however, need to be improved. There is a need to include many new machine learning related references in the topics in the reference list. Here are some references:

# “Scalable Inverse Reinforcement Learning Through Multi-Fidelity Bayesian Optimization” IEEE Transactions on Neural Networks and Learning Systems, 2021.

# "Bayesian Multiple Index Models for Environmental Mixtures." arXiv preprint arXiv:2101.05352 (2021).

# "Optimal Finite-Horizon Perturbation Policy for Inference of Gene Regulatory Networks." IEEE Intelligent Systems (2020).

# "Augmenting machine learning photometric redshifts with Gaussian mixture models." Monthly Notices of the Royal Astronomical Society 498.4 (2020): 5498-5510.

- The format of some of the references is not in standard form. These need to be checked and fixed.

Author Response

We thank the reviewer. As suggested, we reformatted the bibliography based on format established for IJERPH, according to their Endnote formatting file that is available on their site.

Our revision added text to discussion machine learning methods that handle more high dimensional data, which we hope is responsive to the reviewer's concerns. We believe the specific citations provided are not as appealing to the environmental epidemiology audience, which is why they have not been added to the manuscript. We also note that some are, for now, not peer reviewed from what we are able to tell.